# Effector-dependent structural transformation of a crystalline framework with allosteric effects on molecular recognition ability

Ryunosuke Hayashi[1], Shohei Tashiro [1] ✉, Masahiro Asakura[1], Shinya Mitsui[1] & Mitsuhiko Shionoya [1] ✉

Structurally flexible porous crystals that combine high regularity and stimuli responsiveness have received attracted attention in connection with natural allostery found in regulatory systems of activity and function in biological systems. Porous crystals with molecular recognition sites in the inner pores are particularly promising for achieving elaborate functional control, where the local binding of effectors triggers their distortion to propagate throughout the structure. Here we report that the structure of a porous molecular crystal can be allosterically controlled by local adsorption of effectors within low-symmetry nanochannels with multiple molecular recognition sites. The exchange of effectors at the allosteric site triggers diverse conversion of the framework structure in an effector-dependent manner. In conjunction with the structural conversion, it is also possible to switch the molecular affinity at different recognition sites. These results may provide a guideline for the development of supramolecular materials with flexible and highly-ordered three-dimensional structures for biological applications.

Allostery[1] found in proteins and other biological macromolecules is an essential mechanism for the precise control of biological activities. For example, in the case of allosteric enzymes, effector molecules bind to a local site other than the active centre and alter the conformation of the enzyme, thereby modulating the equilibrium of the enzyme-substrate complex formation and the catalytic activity of the enzyme. Inspired by such elaborate biological systems, the development of synthetic host compounds with structural flexibility, stimulus responsiveness and associated allosteric cooperativity has attracted much interest in supramolecular chemistry. Until now, allosteric control has been achieved with discrete synthetic molecules[2,3], such as macrocyclic receptors and supramolecular metal complexes. In these systems, effector molecules bind to specific sites to induce conformational changes of the host compound, resulting in remote control of molecular recognition[4–9] and catalysis[10–13] at different sites in the process of metal ion inclusion or catalysis at the metal centre.

On the other hand, the emergence of metal-organic frameworks (MOFs)[14,15], covalent organic frameworks (COFs)[16] and even hydrogen-bonded organic frameworks (HOFs)[17] has led to the development of structurally flexible porous crystalline hosts[18–25]. Such flexible porous crystals are regarded as promising materials for many applications such as molecular separation[26–31], transport[32–34], sensing[35,36], catalysis[37–39] and crystalline sponge method[40,41] taking advantage of the synergistic effects of porosity and structural flexibility. The structural transformation of these crystalline hosts is generally induced by external stimuli such as temperature, gas pressure, light and solvents. Furthermore, structural transformation by molecular recognition processes[42–46], such as enzymes, is currently being investigated and, similar to the structural diversity of biopolymers, various transformation modes can be derived depending on the specific host-guest interaction patterns. For example, Rosseinsky et al. determined nine different crystal structures of a Zn-peptide MOF by structural control, depending on the spatial uptake of guest molecules, to switch the

[1]Department of Chemistry, Graduate School of Science, The University of Tokyo, 7-3-1 Hongo, Bunkyo-ku, Tokyo 113-0033, Japan.
✉e-mail: tashiro@chem.s.u-tokyo.ac.jp; shionoya@chem.s.u-tokyo.ac.jp

adsorption ability of other small molecules ON and OFF[44]. In such conventional guest-induced transformations, most of the intrinsic active space is occupied by the guest molecules because the crystal structure is deformed to match the shape of the guest molecules incorporated throughout the pore. In other words, it remains challenging to design a crystalline host that allows precise molecular recognition at independent allosteric sites, such as proteins, and remote control of structures and functions without filling the active space (Fig. 1a).

Our group has previously reported a porous metal-macrocycle framework (MMF) constructed by co-crystallisation of equimolar amounts of four stereoisomeric $Pd^{II}$-macrocycles ($C_3$-symmetric (*P/M*)-*syn*, $C_1$-symmetric (*P/M*)-*anti*) via hydrogen bonding and $Pd^{II}$–$Pd^{II}$ interactions (Fig. 1b)[47,48]. We have already reported that the built-in one-dimensional nanochannel ($1.9 \times 1.4\ nm^2$ cross-section) can function as a reaction field for acid-catalysis and photocatalysis[49,50]. The most striking structural feature of the MMF channel is the presence of multiple molecular recognition sites on the less symmetric inner wall, which is attributed to the internal or interstitial voids in the $Pd^{II}$-macrocycles. Therefore, guest molecules with various structures and functional groups can be site-selectively adsorbed and arranged at specific sites within a single nanochannel[51,52]. Therefore, assuming that the rearrangement of $Pd^{II}$-macrocycles occurs by site-selective adsorption of specific effector molecules, the entire crystal structure and inherent functions of MMF may be controlled allosterically without filling the channel with guest molecules. As a proof-of-concept, we report here that a variety of crystal structural transformations occur depending on the local adsorption of effector molecules and that it is possible to control the molecular recognition ability of different sites in a coordinated manner (Fig. 1a). Isolated molecular recognition sites of MMF function as allosteric sites, reversibly adsorbing effectors and inducing specific crystal-to-crystal structural transformation (Fig. 1c). Furthermore, rearrangement of the crystal components allowed allosteric control of the ability to recognise molecules at distant sites within a single nanochannel. The results of this work are highly original in that the allostery observed in the relatively simple synthetic hosts is fused with the space-specific functionality found in the crystalline framework. These results suggest that macrocycle-based molecular crystals have high potential in the search for supramolecular materials with biomimetic structural flexibility.

## Results and discussion

Multiple acetonitrile (MeCN) molecules are adsorbed on the inner walls of the nanochannels of MMF crystallised in MeCN. One of them is accommodated in each specific interstitial binding site surrounded by four $Pd^{II}$-macrocycles at the bottom corner of the channel via multipoint non-covalent interactions (Fig. 1c). Since these interstitial sites are located at the nodes of the vertical and horizontal columns of the framework, it is expected that the crystal structure can be deformed vertically and horizontally by propagating the strain generated here. Therefore, MMF crystals were soaked in various polar organic solvents to induce the structural transformation of $Pd^{II}$-macrocycles by replacing MeCN with appropriate molecules.

First, MMF crystals were soaked in several ethers at 20 °C for 1 day, then transferred to a glass capillary and powder X-ray diffraction (PXRD) was measured at room temperature. The resulting changes in the diffraction peaks suggested a peculiar structural transformation (Fig. 2). For instance, in dimethoxyethane (DME), the 100 and 110 diffraction peaks were observed to shift significantly from $2\theta = 4.40$, 4.70° to $2\theta = 3.84$, 4.16°. These peak shifts to low-angle regions suggest anisotropic structural extensions along the crystallographic *a*-axis. On the other hand, no peak shifts were detected in other linear ether analogues. For cyclic ethers, a distinctive change in diffraction pattern was observed in 1,4-dioxane. In this case, the diffraction peaks derived from other planes exhibited relatively large shifts, e.g. the 040 peaks

were found to shift from $2\theta = 6.56°$ to 6.36°. No comparable shifts were observed for the other cyclic ethers. In other words, a crystal structural transformation highly specific to the ether structure was demonstrated.

MMF crystals soaked in ethers were then analysed by single-crystal X-ray diffraction (SCXRD) at –180 °C. The obtained crystal structures were consistent with the above-mentioned PXRD results and showed a peculiar transformation (Fig. 3). In DME, the *a*-axis cell parameters increased by 12.5% with a 29.3% increase in void space volume (MMF⊃DME: $a = 22.0352(2)$ Å, MMF⊃MeCN: $a = 19.5910(8)$ Å), and in 1,4-dioxane, the *b*-axis cell parameters increased by 4.3% with a 13.3% increase in void space volume (MMF⊃1,4-dioxane: $b = 53.9518(12)$ Å, MMF⊃MeCN: $b = 51.730(3)$ Å). In both cases, the conformation of the $Pd^{II}$-macrocycles did not change upon transformation, indicating that the structural transformation was due to differences in their relative positions in the assembled structure. On the interstitial site, the diether molecule was adsorbed via multipoint hydrogen bonds (NH···O, CH···O, CH···Cl) with multiple $Pd^{II}$-macrocycles, and deformed the pocket according to its shape and binding mode. DME was accommodated in the gaps among the *syn*-isomers constituting the vertical 2D-network, while 1,4-dioxane was interposed between the 2D-network of *syn*-isomers and the *anti*-isomer constituting the horizontal pillar. As a result of the deformation of this pocket, these diethers allow the structure to extend anisotropically in a certain direction (Supplementary Figs. 2, 4). Dissociation and reorganisation of intermolecular interactions between *syn*-isomers were observed, especially in DME, accompanied by the formation of new hydrogen bonds and $Pd^{II}$–$Pd^{II}$ interactions. Other linear ethers were also introduced into the nanochannel but were not incorporated into the interstitial sites and did not induce changes in the framework structure (Supplementary Figs. 6, 8, 10, 14). Cyclic 1,3-dioxolane was incorporated into the interstitial sites as well as 1,4-dioxane, but due to its smaller molecular size, only a slight structural extension was observed (Supplementary Fig. 12). These structural transformations were also confirmed to be reversible processes that return to the original structure by re-soaking in acetonitrile (Supplementary Figs. 102, 106, 107). In summary, the bottom-corner interstitial sites function as adaptive and specific allosteric sites, triggering a reversible structural transformation of the entire MMF by effector exchange. Therefore, multi-directional anisotropic extension depending on the effectors was realised in a crystal-to-crystal manner.

Further examination revealed that allosteric structural transformation can be induced diversely by recognising various effectors. Similar soaking experiments showed that aliphatic/aromatic compounds with polar functional groups such as ether, hydroxy, carbonyl, nitrile and nitro-groups adsorbed on the same allosteric site, resulting in specific structural changes of MMF. For instance, in tetraethylene glycol dimethyl ether (tetraglyme), the *a*-axis cell parameter increased by 15.2% (MMF⊃tetraglyme: $a = 22.5704(5)$ Å) in SCXRD analysis, confirming structural extension (Fig. 4a). Tetraglyme was found to adsorb across two allosteric sites in adjacent channels as an effector, penetrating newly formed pinholes in the reconstructed 2D-network of *syn*-isomers and forming interlaced structures after synthesis. 1,1-Bis(hydroxymethyl)cyclopropane (BHC) with a three-membered ring also adsorbed in the gaps between the *syn*-isomers (Fig. 4b). One OH group bridged the $Pd^{II}$-macrocycles to form a new $Pd^{II}$–$Pd^{II}$ interaction, while the opposite OH group bound to the NH of another complex. This binding mode significantly pushed up the $Pd^{II}$-macrocycles, with a marked increase in the *a*-axis cell parameter of 20.7% (Supplementary Fig. 40). Secondary alcohols were accommodated via hydrogen bonds between the OH group and the NH group of one complex. In this case, continuous changes in the extension rate with side chain length were observed, with (*rac*)-1-phenylethanol (PEA) increasing cell parameters by 5.2% in the *a*-axis direction and 5.8% in the *b*-axis direction (Supplementary Figs. 57, 67). Alternatively, SCXRD measurements of MMF

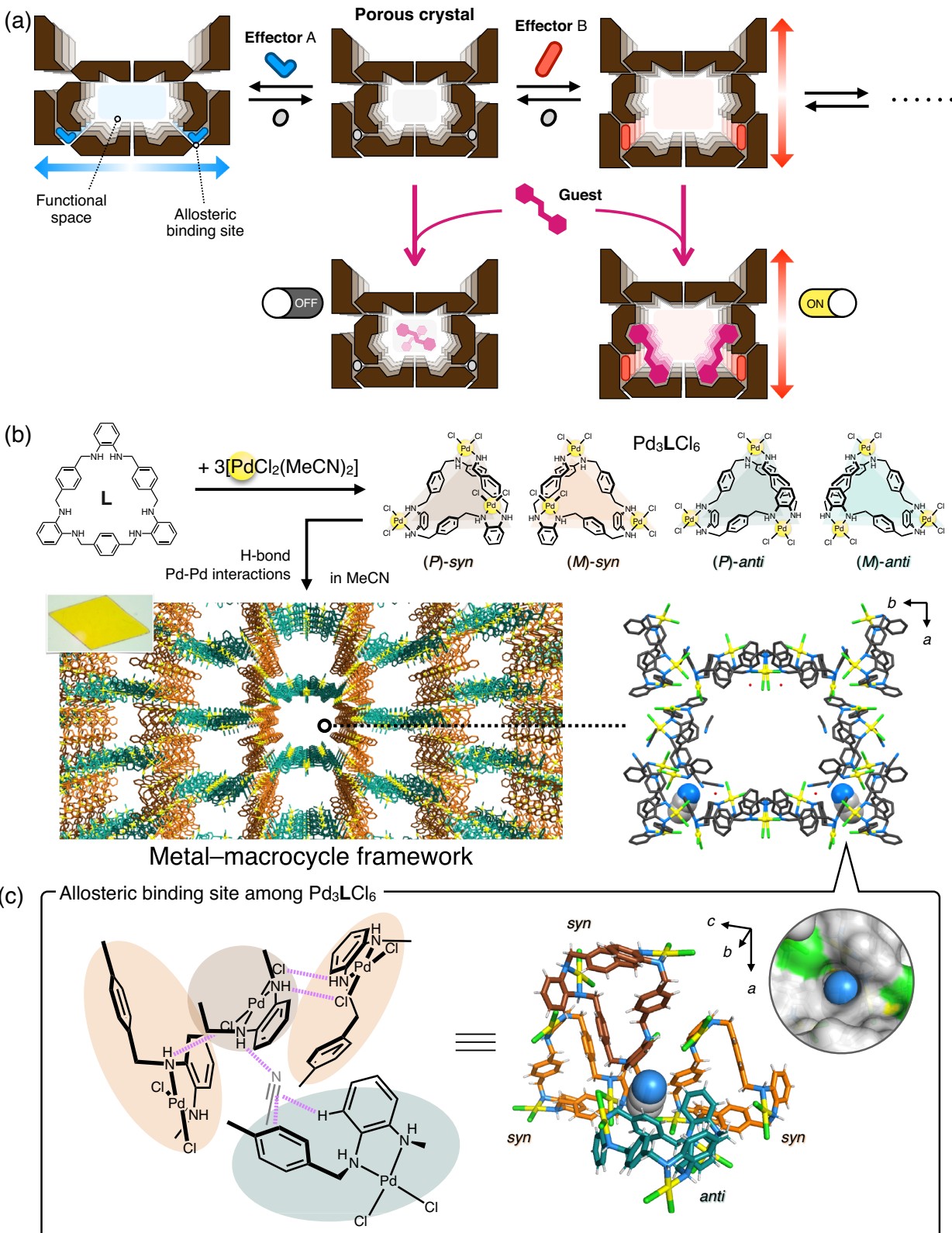

**Fig. 1 | Concept of this study and crystal structure of MMF. a** Schematic diagram of a flexible crystalline host whose structure and function can be remotely controlled by precise molecular recognition at independent allosteric sites. **b** Crystal structure and microscopic image of an MMF crystal co-crystallised with four stereoisomers of Pd$^{II}$-macrocycle. **c** Chemical structure of the interstitial binding site at the bottom corner of the channel that serves as an allosteric site to accommodate effectors. The magenta dotted lines indicate non-covalent interactions such as hydrogen bonds, CH–π and π–π interactions. The (*P*)-*syn*, (*M*)-*syn*, (*P*)-*anti* and (*M*)-*anti* isomers of the Pd$^{II}$-macrocycle are shown in brown, orange, teal and turquoise, respectively.

crystals soaked in volatile solvents such as diethyl ether (Et$_2$O) at −180 °C showed structural contraction associated with allosteric site closure, with a 13.5% decrease in void volume (Supplementary Fig. 75).

The rate of structural transformation was then estimated: for small effectors such as DME, SCXRD analysis confirmed that structural transformation occurred after only 50 s of soaking at 20 °C, and effector molecules were accommodated in the allosteric sites in equilibrium (Supplementary Fig. 98). Many cracks were observed in the crystals due to abrupt transformation. On the other hand, more rigid and bulkier effectors, such as aromatics, require a longer time.

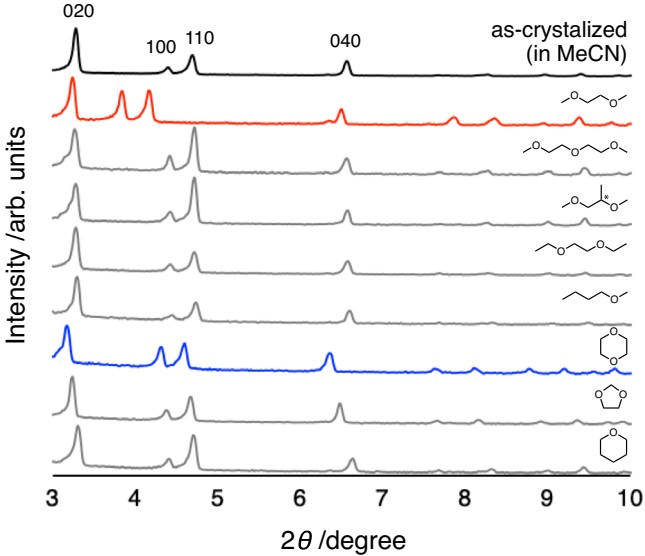

**Fig. 2 | Powder X-ray diffraction (PXRD) patterns of MMF soaked in several ethers.** PXRD profiles (rt, CuKα) for MMF crystals packed in a glass capillary with organic solvents: acetonitrile (black), 1,2-dimethoxyethane (red), diethylene glycol dimethyl ether, (*rac*)−1,2-dimethoxypropane, 1,2-diethoxyethane, methyl *n*-butyl ether, 1,4-dioxane (blue), 1,3-dioxolane and tetrahydropyran.

For example, in acetophenone (ACP), MMF showed no structural changes after one day of soaking at 20 °C (Supplementary Fig. 100). When the crystals were heated at 100 °C for 3 h, structural extension in the *a*-axis direction was confirmed in PXRD analysis at room temperature (Supplementary Fig. 109). The crystal structure obtained by SCXRD showed a 15.7% increase in the *a*-axis cell parameter and the formation of a 2D-network of *syn*-isomers similar to that in DME (Supplementary Fig. 44). ACP was incorporated into the allosteric sites via multipoint interactions such as NH···O hydrogen bonds and CH−π interactions. Given that large crystal thermal fluctuations are required for rigid and bulky effectors to adsorb to the allosteric sites, the higher activation energy of effector accommodation may be responsible for the slow transformation.

In the above experiments, the effector molecules were used as neat oils, but structural transformations were also observed in diluted solutions under optimised conditions. For example, when MMF was soaked in a solution of resorcinol in 1,2,3,4-tetrahydronaphthalene:acetone = 9:1 (vol:vol) (500 mM) at 20 °C, an extended structure was observed with high phase purity by PXRD analysis (Supplementary Fig. 112). Accommodation of resorcinol at allosteric sites was confirmed by SCXRD analysis, with a 15.2% increase in the *a*-axis cell parameter (Supplementary Fig. 52).

Furthermore, the application of benzyl alcohol and its derivatives as effectors enabled stepwise structural extension. Soaking of MMF in benzyl alcohol (BnOH) at 20 °C for 1 day increased the *a*-axis cell parameter by 4.3% and void volume by 9.4%, indicating moderate structural extension, according to PXRD (Supplementary Fig. 116) and SCXRD analyses (Fig. 4c). BnOH was accommodated between the two *syn*-complexes on the allosteric sites due to multipoint non-covalent interactions, forming additional π−Cl interactions between the Pd$^{II}$-macrocycles. Further annealing at 125 °C resulted in further structural transformations, as indicated by PXRD analysis. The corresponding crystal structure obtained by SCXRD exhibited a significant structural extension, with a 15.7% increase in the *a*-axis cell parameter and a 43.6% increase in the void volume compared to the as-crystalised one (Fig. 4c and Supplementary Fig. 78). This indicates that, as a result of the conformational change of BnOH, the *syn*-complexes rearranged their interactions and formed a network similar to that in DME. Focusing on

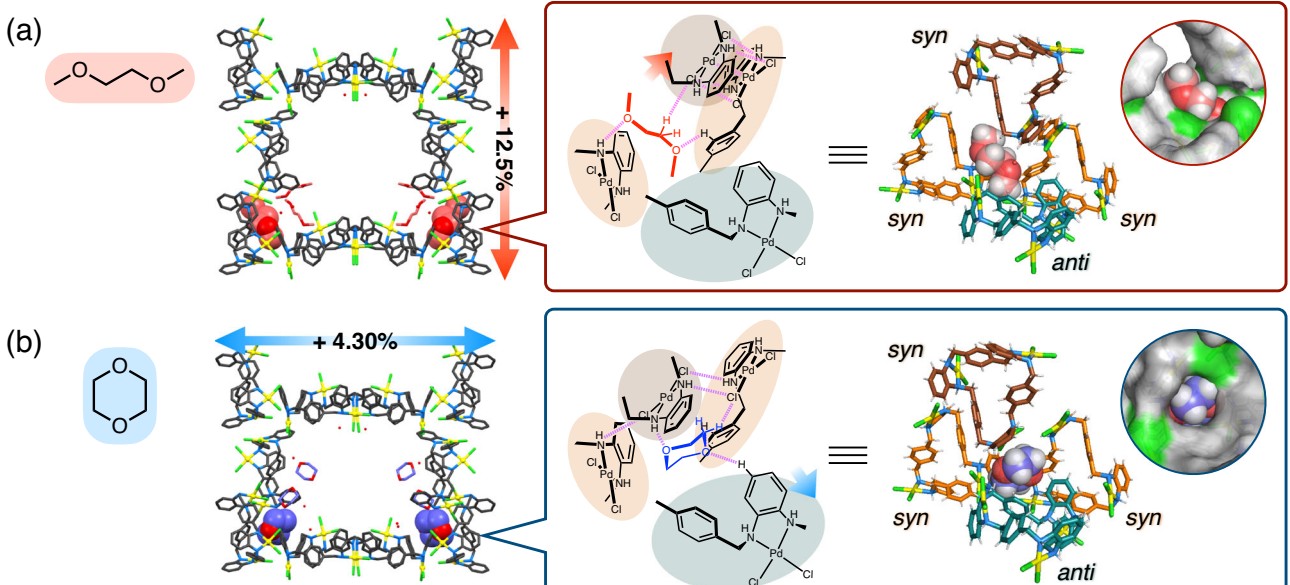

**Fig. 3 | Crystal structures of the unit-space and the allosteric binding sites of the structurally extended MMF with effectors. a** 1,2-Dimethoxyethane (red) and **b** 1,4-dioxane (blue). The magenta dotted lines show non-covalent interactions (NH···O, CH···O, NH···Cl and CH···Cl) between an effector and a Pd$^{II}$-macrocycle or between two Pd$^{II}$-macrocycles. In the structures of the allosteric binding site, the (*P*)-*syn*, (*M*)-*syn*, (*P*)-*anti* and (*M*)-*anti* isomers of the Pd$^{II}$-macrocycle are shown in brown, orange, teal and turquoise, respectively.

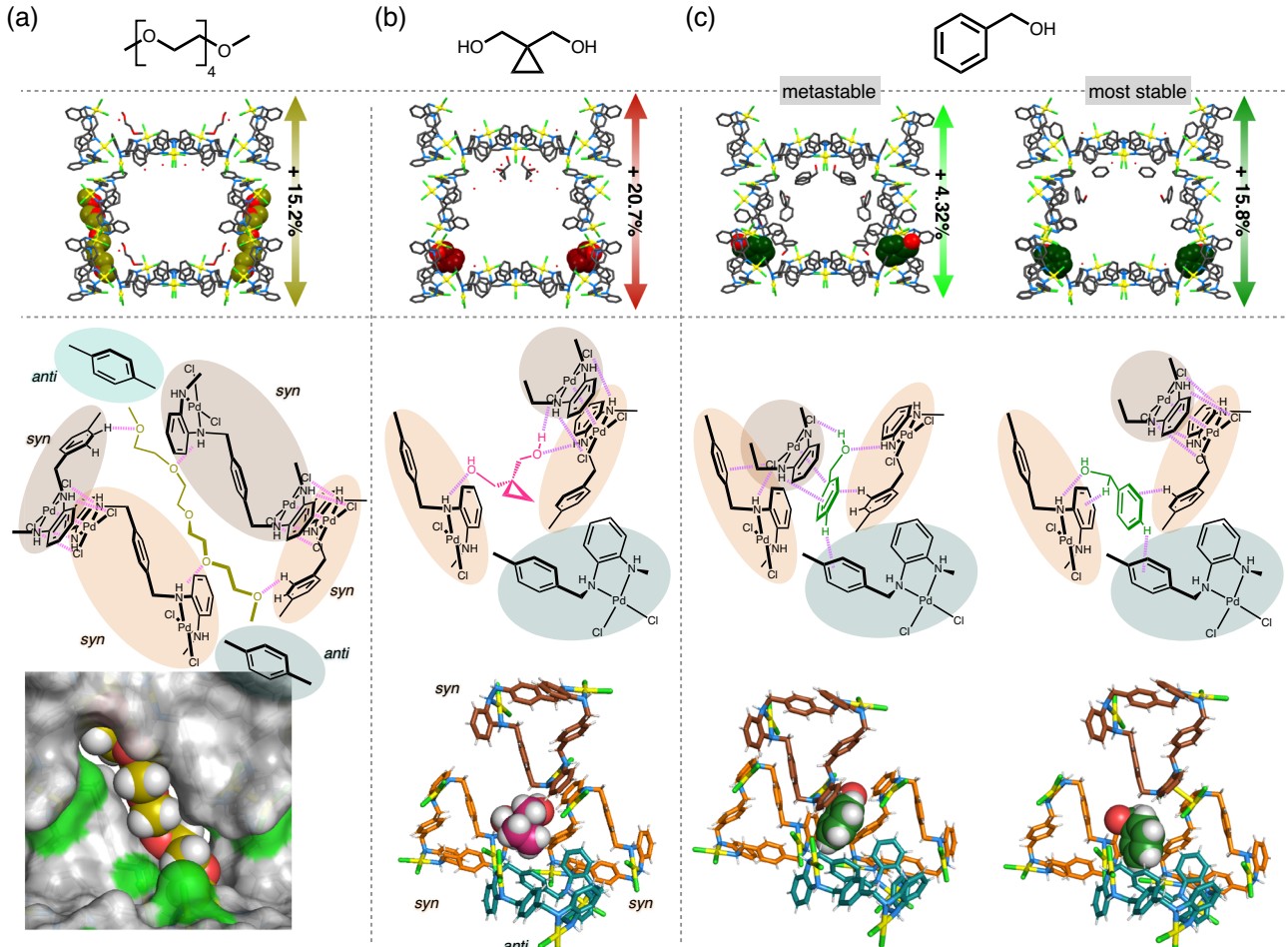

**Fig. 4 | MMF unit-space structure and interaction patterns around its allosteric site accommodating an effector. a** Tetraethylene glycol dimethyl ether (yellow), **b** 1,1-bis(hydroxymethyl)cyclopropane (pink) and **c** benzyl alcohol (green, left; metastable crystal structure observed in the first step, right; the most stable crystal structure obtained in the second step). The magenta dotted lines represent non-covalent interactions between an effector and a Pd$^{II}$-macrocycle or between two Pd$^{II}$-macrocycles. In the structures of the allosteric binding site, the (*P*)-*syn*, (*M*)-*syn*, (*P*)-*anti* and (*M*)-*anti* isomers of the Pd$^{II}$-macrocycle are shown in brown, orange, teal and turquoise, respectively.

the intermolecular interactions, this phase transition can be rationalised as follows. In the first step, partial recombination of the inherent interactions between Pd$^{II}$-macrocycles resulted in a moderately extended structure as a metastable state due to effector accommodation. Once the large activation barrier was overcome in the second step, its bonding network was extensively reorganised, resulting in the most stable or significantly extended structure with more effective intermolecular interactions.

So far, more than 40 effector molecules have been discovered, and MMF structure transformations specific to each effector have been observed. To investigate the correlation of various structural transformation modes, MMF crystal structures incorporating effectors were mathematically characterised by principal component analysis with intermolecular N/C···Cl atomic distances of the Pd$^{II}$-macrocycles as variables (Fig. 5a and Supplementary Fig. 103). This result suggests that the overall structures can be classified into three major clusters I–III (Fig. 5b and Supplementary Fig. 104). Crystal structures belonging to different clusters are based on different 2D-network of *syn*-complexes (Fig. 5c). Cluster I crystals (including MMF as crystallised) consist of both *syn*-1···*syn*-2 and *syn*-2···*syn*-3 hydrogen bonds, whereas cluster II/III are supported by either of them: *syn*-1···*syn*-2 for cluster II or *syn*-2···*syn*-3 for cluster III due to effector intervention. The coexistence of multiple crystal structures of MMF in the mixture of two effectors in different clusters was experimentally confirmed by PXRD analysis. For

instance, in a 0.7:99.3–0.3:99.7 (vol:vol) mixture of MeCN and DME, two different crystal structures of cluster I/III coexisted due to competition for effector accommodation (Supplementary Fig. 125). This suggests that there are significant energy barriers between MMFs characterised by different clusters due to the reconstruction of the 2D-network of the *syn*-complexes. The process of adopting one of the three energetical local minima of MMF crystals can be viewed as conformational selection, in which the energy ranking of multiple conformations of a protein is reordered by complexation with an effector. On the other hand, in the presence of two effectors in the same structural cluster, e.g., MeCN and 1,4-dioxane, the MMF diffraction peaks shifted gradually depending on their mixing ratio (Supplementary Fig. 126). Thus, the MMF crystals in each cluster are likely to adjust the structures to the most stable state by specific interactions with effectors that are stuck in allosteric sites. These adjustment processes are typically associated with stretching and contraction between the vertical *syn*-complexes 2D-network and the *anti*-complexes constituting the horizontal pillar. Such structural misalignment may be analogous to an induced-fit mechanism in which the binding site is continually deformed by interactions with the substrate as the protein forms a complex. In other words, the structural transformation of MMF crystals may be based on a complex mechanism involving both selection and adjustment processes[53].

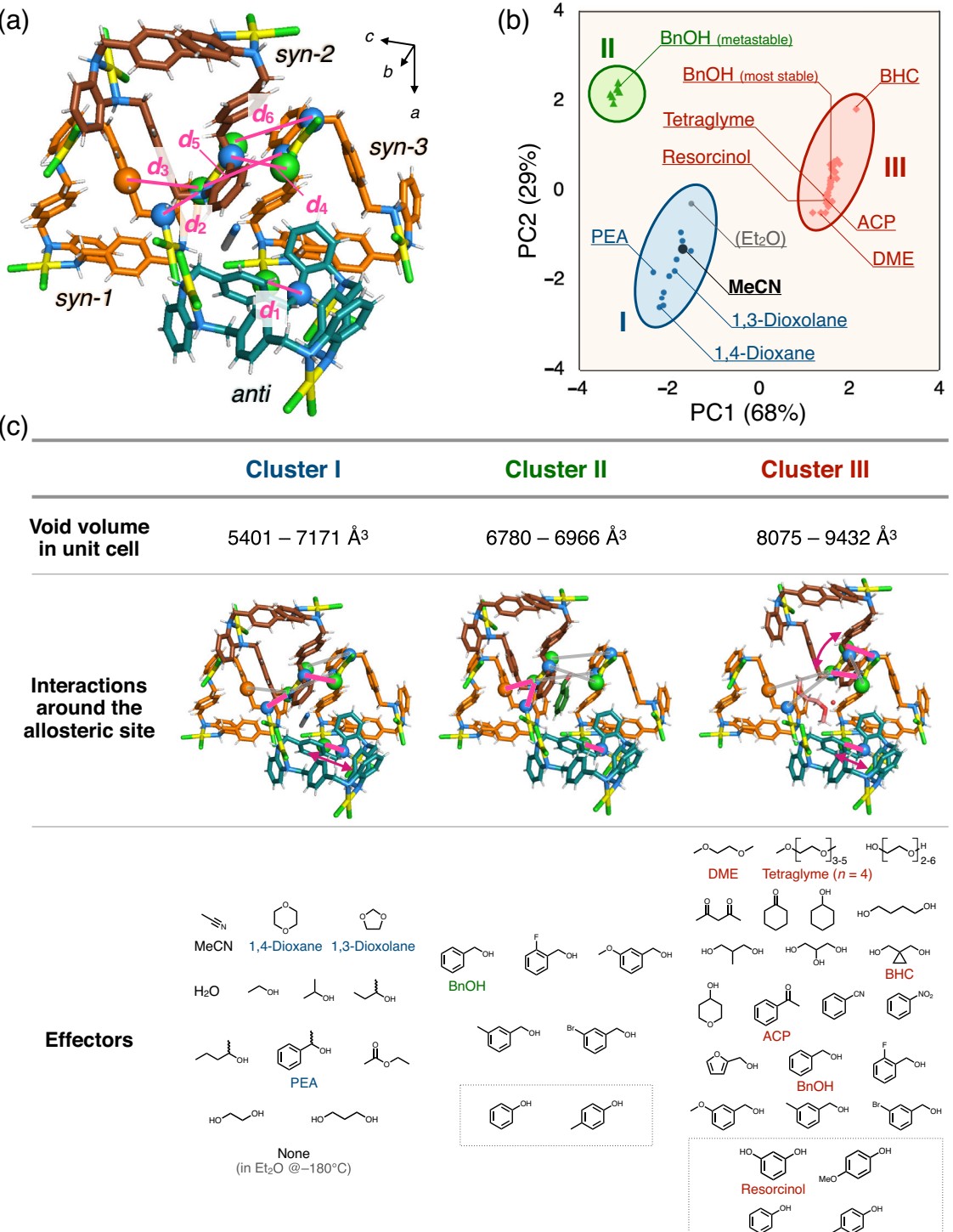

**Fig. 5 | Characterisation of MMF crystal structures by principal component analysis based on interatomic distances in Pd$^{II}$-macrocycles. a** Internuclear distances ($d_1$–$d_6$) of six atomic pairs (N/C⋯Cl) used as variables. **b** PCA score plot of 50 MMF crystal structures with various effectors (variance; PC1 = 68%, PC2 = 29%). Typical structures are labelled with the accommodated effectors, and all structures are categorised into three clusters I–III. **c** The characteristics of three structural clusters: the void volume in the unit-cell, interactions around allosteric sites, and effectors accommodated. The pink lines in (**c**) represent atom pairs that are close enough to interact, while the grey lines represent unbound atom pairs. The pink arrows indicate structural misalignment due to specific interactions with effector molecules in each cluster. All seven effectors in cluster II can also afford cluster III, which is generally the most stable structure. The molecules enclosed by the rectangular dashed box in (**c**) are solid effectors housed in the solution. For solid effectors, the crystal structures obtained in 1,2,3,4-tetrahydronaphthalene solutions were used for PCA analysis.

The structural transformation of MMF was caused by effector molecules binding to independent allosteric sites, leaving all structures with the intrinsic void filled with liquid, amounting to more than 40% of the volume. Therefore, it is possible to cooperatively control the properties of the active space through structural transformation. Herein, we examined the allosteric effects on the molecular arrangement in the channel. MMF crystals before and after the structural transformation were soaked in a guest molecule solution at 20 °C and

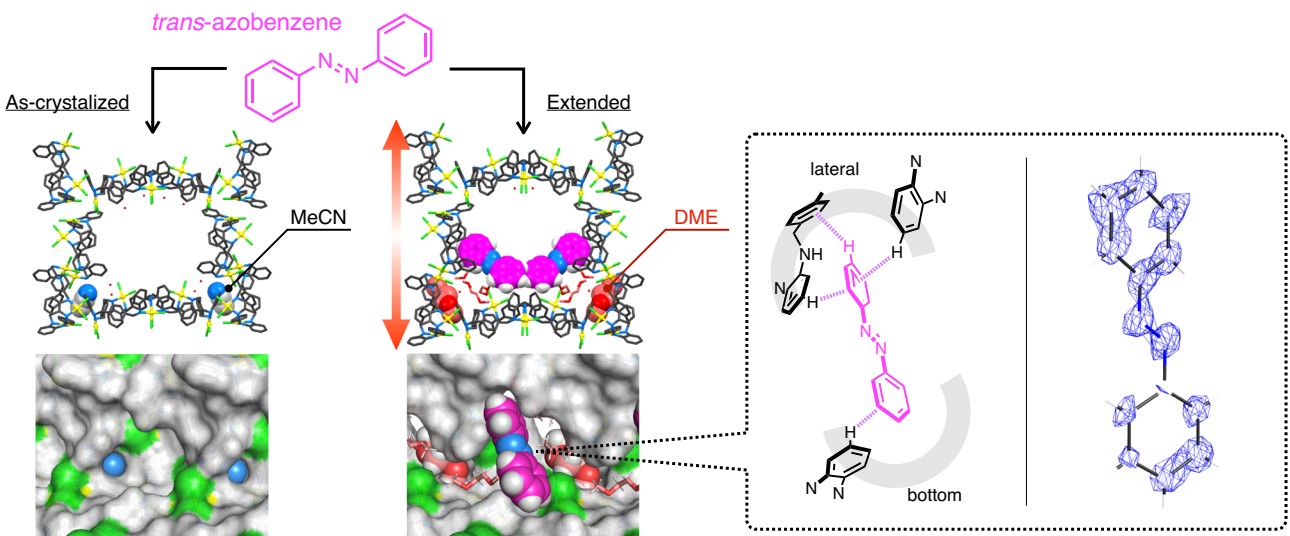

**Fig. 6 | Allosteric effects of structural changes in MMF channels on the molecular recognition ability for *trans*-azobenzene.** Left; the unit-space structure and channel surface structure of as-crystalised MMF (effector: MeCN) soaked in a DME:MeCN = 95:5 (vol:vol) solution of *trans*-azobenzene (1.0 M), right; the unit-space structure, channel surface, host-guest interaction patterns and electron density maps of *trans*-azobenzene adsorbed on a structurally extended MMF (effector: DME) soaked in a DME solution of *trans*-azobenzene (1.0 M). The contour level of the electron density map is 0.7σ. In the chemical and crystal structures, *trans*-azobenzene molecules are shown in pink.

compared by structural analysis using SCXRD. When *trans*-azobenzene (1.0 M) was used as a guest molecule, a positive allosteric effect was observed due to the structural extension induced by the effector molecule, DME (Fig. 6). The guest molecule was not adsorbed on the inner wall of the as-crystalised framework (effector: MeCN)[49], but when the effector was DME, the guest molecule could be adsorbed and arranged on the extended framework due to multipoint CH−π interactions with the Pd[II]-macrocycles on the channel sides and bottom, and a clear electron density distribution was shown (occupancy: 42.6%). In control experiments using a DME:MeCN = 95:5 (vol:vol) solution, the MMF structure remained contracted by preferentially accommodating MeCN as an effector. Even in this case, *trans*-azobenzene was not adsorbed on the wall, suggesting that the effect of solvation on guest adsorption is negligible. [1]H NMR analyses of crystals dissolved in DMSO-$d_6$/DCl showed that *trans*-azobenzene diffused well in both contracted/extended channels (2.0 molecules per contracted unit space and 3.1 molecules per extended unit space, Supplementary Fig. 131). Therefore, the difference in guest adsorption may be attributed to the ability of the guest molecules introduced into the channel to form effective interactions with the inner wall of the MMF. Indeed, the relative positions of the Pd[II]-macrocycles on the channel sides and bottom were significantly altered by the structural transformation, confirming that the as-crystalised framework has no room to form an effective interaction with *trans*-azobenzene. These results indicate that the adsorption of *trans*-azobenzene can be controlled allosterically based on the formation of compatible recognition sites by structural transformation. In addition, allosteric effects on the ability of the channel surface to recognise molecules were observed for other guest molecules. For example, structural extension with 1,4-dioxane had a negative effect on the adsorption of *N*-(benzyloxycarbonyl)-L-serine (Supplementary Fig. 133), which is usually seen at the ceiling site of the as-crystalised framework (effector: MeCN)[52]. The *p*-dibromobenzene that was immobilised on the sidewalls of the as-crystalised framework (effector: MeCN)[47] was instead adsorbed in the newly formed gap on the bottom wall after structural extension with DME (Supplementary Fig. 138). Allosteric control of molecular arrangement in porous spaces could lead not only to typical applications such as molecular transport and property control, but also to mimic sophisticated activity control such as protein feedback mechanisms.

In conclusion, we have achieved post-synthetic structural control of MMF crystals with a low-symmetric nanochannel based on reversible effector binding to specific sites in the channel. Complexation with various effectors induces specific structural transformations such as anisotropic extension, framework penetration and stepwise changes through a process of selection and adjustment. In addition, allosteric cooperative actions involving molecular recognition processes in isolated nanochannel spaces were also demonstrated. This study is expected to be the leading edge in the development of bio-inspired supramolecular materials with precise molecular control capabilities and a high degree of structural flexibility in specific nanospaces, leading to elaborate porous functions such as enzyme-mimetic catalysis and more effective crystalline sponge methods.

## Methods
### Synthesis of MMF crystals
The hexaamine macrocyclic ligand **L** was synthesised by a modified procedure according to literature reported previously[47]. Terephthalaldehyde (6.68 g, 49.8 mmol) dissolved in dry tetrahydrofuran (THF) (45 mL) was added to *o*-phenylenediamine (5.43 g, 50.3 mmol) dissolved in dry THF (24 mL) at room temperature under an Ar atmosphere. After stirring at 50 °C for 3 h, the mixture was evaporated until the volume was reduced by about half, dried with anhydrous $Na_2SO_4$ and washed with chloroform. The resultant solution was concentrated under reduced pressure, redissolved in THF (20 mL) and concentrated again. After drying under reduced pressure along with a desiccant, the crude material was suspended in dry THF (215 mL) and slowly added to $LiAlH_4$ (1.95 g, 51.3 mmol) suspended in dry THF (43 mL) at room temperature under an Ar atmosphere. Note that when making a suspension of $LiAlH_4$, the $LiAlH_4$ powder must be added slowly to dry THF under an Ar atmosphere to prevent hazards. After refluxing the mixture for 2 h, it was quenched at 0 °C by carefully adding water (2 mL), 15 wt% NaOH aq (2 mL) and water (4 mL) in that order. The mixture was warmed to 60 °C, filtered to remove insoluble material and washed with chloroform. The filtrate was evaporated and separated by silica gel column chromatography using chloroform as the eluent. The separated material was dissolved in chloroform (40 mL), mixed with silica gel (5.3 g) and stirred at room temperature for 1 day to remove coloured impurities. After removal of the silica gel

by filtration, the filtrate was concentrated and recrystallised with dichloromethane and acetonitrile to afford **L** as pale-yellow crystals (0.569 g, 0.902 mmol, 5.4%).

MMF crystals were synthesised by a modified procedure according to literature reported previously[47]. Ligand **L** (10.1 mg, 16.0 μmol) was dissolved in acetonitrile (57 mL) at 80 °C and an acetonitrile solution of $PdCl_2(CH_3CN)_2$ (20 mM, 2.55 mL, 51.0 μmol) was added. The mixture was immediately filtered to remove dust, and the filtrate was allowed to stand at room temperature for several days to yield yellow platelet crystals of MMF. The supernatant was replaced several times with acetonitrile to prevent drying and then stored in acetonitrile.

## Data availability
The photographs of crystals, and powder X-ray diffraction data generated in this study are provided in the main text and the Supplementary Information. The X-ray crystallographic coordinates for structures reported in this study have been deposited at the Cambridge Crystallographic Data Centre (CCDC), under deposition numbers CCDC 2223883-2223889, 2223897-2223942 and 2223947-2223955. These data can be obtained free of charge from the Cambridge Crystallographic Data Centre via www.ccdc.cam.ac.uk/data_request/cif.

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

## Acknowledgements

This research was supported by the JSPS KAKENHI, grant number JP16H06509 (Coordination Asymmetry) (M.S.) and JP18H04502 (Soft Crystals) (S.T.), and partly supported by JSPS KAKENHI, grant number JP22H04522 (Aquatic Functional Materials) (S.T.), JST PRESTO, grant number JPMJPR22A8 (S.T.), Iketani Science and Technology Foundation (S.T.) and Fujimori Science and Technology Foundation (S.T.). R.H. thanks the Program of Excellence in Photon Science (XPS) offered by the University of Tokyo. The PXRD measurement was supported by Advanced Research Infrastructure for Materials and Nanotechnology in Japan (ARIM) of the Ministry of Education, Culture, Sports, Science and Technology (MEXT), grant number JPMXP1222UT0057. We are grateful to Prof. Eiji Nishibori (Tsukuba Univ.) for the PXRD measurement in the preliminary phase of this study.

## Author contributions

R.H. performed almost all the experiments and analyses. M.A. and S.M. conducted the initial phase of the experiments. R.H., S.T. and M.S. designed the project and prepared the manuscript.

## Competing interests

The authors declare no competing interests.
