## [Peer Review File · Nature Communications]

Effector-dependent structural transformation of a crystalline framework with allosteric effects on molecular recognition abilityReviewers' Comments:

Reviewer #1:

Remarks to the Author:

The manuscript by Tashiro and Shionoya titled "Effector-dependent structural transformation of a crystalline framework with allosteric effects on molecular recognition ability" is a very comprehensive crystal engineering study of a porous adaptable MOF from a simple PdII-macrocycles. These MOF have sizable pores which can trap various small solvent and guest molecules.

The work reports 62 single crystal structures of the mentioned MOF with various solvent and guest combinations.

The work, in a sense, tries to emulate the Crystalline Sponge Method (CSM) by Makoto Fujita, but from a more flexible MOF. Surprisingly the author do not cite the Fujita CSM, as also his MOFs are (somewhat) adaptable to changed by an effector (as the authors name other components).

Eventhough the findings are interesting there is not really interesting target molecules that have been "recognized". As the effectors (which could also be called co-formers) the authors have used are simple ethers, alcohols or ketones, but they are all either non-chiral or racemic (Figure 5). The work focuses more on probing the resulting ternary systems, but no real guests (as is the case with CSM), in addition to the effector, have been used.

In this respect this work feels like a well done and extensive, but still rather, routine crystal engineering study that does not report anything really new and unprecedented on porous MOFs.

This work is better suited on a crystal engineering journals, e.g. Crystal Growth and Design.

Reviewer #2:

Remarks to the Author:

The work by Hayashi is an elegant study of some Pd macrocycles with inner porosity. The molecular cages respond with a structural change upon exposure to various solvent molecules, here termed effectors. In the presence of larger molecules the effector can trigger the pore opening (or pore space change) and a larger molecule like azobenzene is adsorbed. The study encompasses the investigations of 40 different effectors and various guest molecules. The study relies on detailed single crystal work and is very elegant and elaborate.

All findings are substantiated by spectroscopy and/or diffraction experiments.

This work is an important step towards engineering materials with highly specific recognition mechanisms and is a valuable contribution to Nat. Commun.

Reviewer #3:

Remarks to the Author:

The manuscript by Shionoya et al. describes structural changes in a previously reported extended framework, formed from packing of a metallocyclic assembly, with 1D pores. Single-crystal-to-single-crystal changes are observed upon soaking the framework in a range of ethers/alcohols, referred to as "effectors" – over 40 have been examined in this work!

Subsequently the authors have investigated how the structural changes induced by the effector molecules alter guest-binding properties within the 1D pores/channels. Examples of both positive and negative effects on guest-binding, in comparison to the "as-synthesised" framework, are reported.

This is a really nice piece of work that demonstrates the use of an allosteric binding mechanisms to alter the conformation of the framework's main binding channels and subsequently guest-binding at this site. The conclusions are backed-up by extensive single-crystal X-ray diffraction studies. Although the ability of this material to take up a variety of guests has been explored previously, I believe this is a significant and interesting enough advancement that it will be of great interest to various communities in the chemical sciences (frameworks, adaptable materials, supramolecular hosts) and thus I recommend publication in Nature Communications.

Reviewer #1's comments and our responses

The manuscript by Tashiro and Shionoya titled "Effector-dependent structural transformation of a crystalline framework with allosteric effects on molecular recognition ability" is a very comprehensive crystal engineering study of a porous adaptable MOF from a simple PdII-macrocycles. These MOF have sizable pores which can trap various small solvent and guest molecules.

The work reports 62 single crystal structures of the mentioned MOF with various solvent and guest combinations.

The work, in a sense, tries to emulate the Crystalline Sponge Method (CSM) by Makoto Fujita, but from a more flexible MOF. Surprisingly the author do not cite the Fujita CSM, as also his MOFs are (somewhat) adaptable to be changed by an effector (as the authors name other components).

Even though the findings are interesting there is not really interesting target molecules that have been "recognized". As the effectors (which could also be called co-formers) the authors have used are simple ethers, alcohols or ketones, but they are all either non-chiral or racemic (Figure 5). The work focuses more on probing the resulting ternary systems, but no real guests (as is the case with CSM), in addition to the effector, have been used.

In this respect this work feels like a well done and extensive, but still rather, routine crystal engineering study that does not report anything really new and unprecedented on porous MOFs.

This work is better suited on a crystal engineering journals, e.g. Crystal Growth and Design.

Our responses:

Thank you for your comments. We understand that your suggestion is based on the recognition that the purpose of this study is to develop a crystalline sponge method (CSM). However, as stated in this manuscript, the goal of this study is not to develop CSMs, but to achieve allosteric control of molecular recognition behavior through enzyme-mimetic structural control of the crystalline backbone by local adsorption of guest molecules. Therefore, the points made by this reviewer's point seems a bit off the mark. On the other hand, the proposed references are important to this study because of their relevance to the outlook of this study. One of the flexible MOFs reported by Fujita et al. has already been cited as reference 22, but not for CSM. Therefore, we added a recent review on CSM by Fujita et al. and a recent paper about CSM with a flexible MOF as new references 40 and 41, respectively. Moreover, to make it clearer that the CSM is not our objective, but one of our future perspectives, the last part of the conclusion has been modified as follows. We thank you for pointing this out.

The second paragraph of the introduction part (yellow parts are added.)

On the other hand, the emergence of metal-organic frameworks (MOFs)^{14,15}, covalent organic frameworks (COFs)¹⁶ and even hydrogen-bonded organic frameworks (HOFs)¹⁷ has led to the development of structurally-flexible porous crystalline hosts^{18–25}. Such flexible porous crystals are regarded as promising materials for many applications such as molecular separation^{26–31}, transport^{32–34}, sensing^{35,36}, catalysis^{37–39} and crystalline sponge method^{40,41} taking advantage of the synergistic effects of porosity and structural flexibility.

The last paragraph of this manuscript (yellow parts are added.)

In conclusion, we have achieved post-synthetic structural control of MMF crystals with a low-symmetric nanochannel based on reversible effector binding to specific sites in the channel. Complexation with various effectors induces specific structural transformation such as anisotropic extension, framework penetration and stepwise changes through a process of selection and adjustment. In addition, allosteric cooperative actions involving molecular recognition processes in isolated nanochannel spaces were also demonstrated. This study is expected to be the leading edge in the development of bio-inspired supramolecular materials with precise molecular control capabilities and a high degree of structural flexibility in specific nanospaces, leading to elaborate porous functions such as enzyme-mimetic catalysis and more effective crystalline sponge methods.

The References section (yellow parts were added.)

40. Zigon, N., Duplan, V., Wada, N. & Fujita, M. Crystalline Sponge Method: X-ray Structure Analysis of Small Molecules by Post-Orientation within Porous Crystals—Principle and Proof-of-Concept Studies, *Angew. Chem. Int. Ed.* **60**, 25204–25222 (2021).

41. Qin, J.-S., Yuan, S., Alsalme, A. & Zhou, H.-C. Flexible Zirconium MOF as the Crystalline Sponge for Coordinative Alignment of Dicarboxylates, *ACS Appl. Mater. Interfaces* **9**, 33408–33412 (2017).

Reviewer #2's comments and our responses

The work by Hayashi is an elegant study of some Pd macrocycles with inner porosity. The molecular cages respond with a structural change upon exposure to various solvent molecules, here termed effectors. In the presence of larger molecules the effector can trigger the pore opening (or pore space change) and a larger molecule like azobenzene is adsorbed. The study encompasses the investigations of 40 different effectors and various guest molecules. The study relies on detailed single crystal work and is very elegant and elaborate.

All findings are substantiated by spectroscopy and/or diffraction experiments.

This work is an important step towards engineering materials with highly specific recognition mechanisms and is a valuable contribution to Nat. Commun.

Our response:

We would like to thank this reviewer for looking over this manuscript and giving an appropriate assessment of this study. As pointed by the reviewer, we believe that this study becomes an important step towards engineering materials with highly specific recognition mechanisms reminiscent of enzymes.

Reviewer #3's comments and our responses

The manuscript by Shionoya et al. describes structural changes in a previously reported extended framework, formed from packing of a metallocyclic assembly, with 1D pores. Single-crystal-to-single-crystal changes are observed upon soaking the framework in a range of ethers/alcohols, referred to as "effectors" – over 40 have been examined in this work!

Subsequently the authors have investigated how the structural changes induced by the effector molecules alter guest-binding properties within the 1D pores/channels. Examples of both positive and negative effects on guest-binding, in comparison to the "as-synthesised" framework, are reported.

This is a really nice piece of work that demonstrates the use of an allosteric binding mechanisms to alter the conformation of the framework's main binding channels and subsequently guest-binding at this site. The conclusions are backed-up by extensive single-crystal X-ray diffraction studies. Although the ability of this material to take up a variety of guests has been explored previously, I believe this is a significant and interesting enough advancement that it will be of great interest to various communities in the chemical sciences (frameworks, adaptable materials, supramolecular hosts) and thus I recommend publication in Nature Communications.

Our response:

We would like to thank this reviewer for looking over this manuscript and giving an appropriate assessment of this study. The conclusions of this study are based on more than 60 crystal structures determined by SCXRD analysis and are considered sufficiently reliable. In addition, as mentioned by the reviewer, the significance and purpose of this study are quite different from our previous studies, which reported only guest recognition in the MMF.